# The impact of wide step width on lower limb coordination and its variability in individuals with flat feet

Fateme Khorramroo[1], Juha M. Hijmans[2], Seyed Hamed Mousavi[3]*

1 Department of Sport Injuries and Biomechanics, Faculty of Sport Sciences and Health, University of Tehran, Tehran, Iran, 2 Department of Rehabilitation Medicine, University of Groningen, University Medical Center Groningen, Groningen, The Netherlands, 3 Department of Sport Injuries and Biomechanics, Faculty of Sport Sciences and Health, University of Tehran, Tehran, Iran

* musavihamed@ut.ac.ir

## Abstract

Flat foot is a common condition marked by the collapse of the medial longitudinal arch, leading to altered lower limb biomechanics and increased risk of musculoskeletal injuries. We aimed to investigate if wide step width changes the lower limb inter-joint coordination and its variability in flat feet individuals. Twenty flat-footed individuals participated in this cross-sectional study. Lower limb kinematics were assessed by 3-dimensional motion analysis during walking and running on a treadmill with preferred and wide step widths while receiving visual feedback. Inter-joint coordination was quantified using vector-coding for joint angles in the hip, knee, and ankle. Wide walking showed a shift towards proximal joint motion in sagittal ankle-knee coordination during loading response during LR (p=0.006), an In-phase motion in transverse ankle-hip coordination during push-off (p=0.004), and an In-phase pattern in frontal knee-hip coordination during mid-stance (p=0.027). Frontal ankle and transverse knee coordination during push-off changed to In-phase (p=0.003). Wide running significantly shifted frontal ankle-hip coordination towards proximal joint motion during mid-stance (p=0.05). Transverse ankle-hip coordination showed an in-phase pattern in wide conditions during push-off (p=0.044), during LR (p=0.022). Wide walking, significantly increased coordination variability of the sagittal ankle-knee during LR and decreased transverse ankle-hip during push-off. Wide walking significantly increased coordination variability in ankle-knee in sagittal plane during LR (p<0.001). Wide running significantly decreased the coordination variability in the ankle-knee sagittal during LR (p<0.001) and knee-hip sagittal during LR (p=0.007) and push-off (p=0.016). The results showed that wide step width can affect inter-joint coordination during walking/running in flat-footed individuals at certain points. These results should be considered when using a wide step width as a gait retraining method for managing flat-footed individuals.

**Data availability statement:** All relevant data are within the paper and its Supporting Information files.

**Funding:** The author(s) received no specific funding for this work.

**Competing interests:** The authors have declared that no competing interests exist.

## Introduction

Flat foot (pes planus), a prevalent condition (11–29% of 18–25-year-old men and women and 13.6% flexible flat feet [1,2], can be classified as flexible or rigid. Flexible flat foot involves arch collapse during weight-bearing stance [3], reversing upon weight removal [1], and responds well to non-surgical treatments [4], whereas rigid flatfoot is characterized by a stiff, flattened arch in all conditions [5] and typically necessitates surgery [4]. In many flat-footed individuals, the subtalar joint excessively pronates during stance phase of gait and remains pronated without turning to supination early enough during the late stance phase [6], which is inefficient for completing the push-off during gait [7].

Prolonged pronation, particularly caused by excessive rearfoot eversion, commonly seen in flat feet, can lead to compensatory changes in tibial [8] and hip rotation [9], resulting in a greater knee valgus angle due to the interconnected movements between rearfoot inversion/eversion and tibial rotation and low back pain [10]. This excessive rearfoot eversion is a significant risk factor for running-related injuries [11], adversely impacting plantar fascia tension and overall foot biomechanics during gait [12]. These compensatory mechanisms can further disrupt the lower extremities joints coordination [4] and potentially lead to increased plantar fascia tension due to excessive rearfoot pronation [12], may trigger increased stress on the knee. Moreover, coordination between adjacent segments has been implicated in the development of injuries such as iliotibial band syndrome [13].

Coordination variability plays a functional role in controlling strategies within the motor system, enhancing adaptability to motion perturbations and task constraints [14]. Vector coding, along with analyzing four coordination phases, is a common method used to assess joint coordination and relationship of segments [15]. This information can be of assistance in a clinical setting [16]. Previous studies reported differences in kinematic coordination behavior between individuals with pes planus and pes cavus feet [9] as well as between individuals with excessive and normal subtalar pronation [17].

Modifying step width to address rearfoot eversion has emerged as a promising approach beside other interventions; i.e. changing speed, foot progression angle or center of pressure [18], soft tissue mobilization and electrotherapy [19,20]. Evidence indicating that wider step widths can effectively reduce rearfoot eversion during walking and running [18,21]. However, the impact of this modification on lower limb inter-joint coordination has yet to be thoroughly investigated. Understanding how variations in step width influence joint coordination in individuals with flat feet is crucial for elucidating the adaptive mechanisms and developing strategies to optimize gait mechanics in this population. This suggests potential applications in injury prevention and rehabilitation for conditions like patellofemoral pain syndrome or tibial stress injuries [11]. Therefore, the purpose of this study is to examine the effects of wide step width through visual feedback on lower extremity joint coordination and its variability in individuals with flat feet. We hypothesized that wide Step guided by visual feedback would improve lower extremity joint coordination and coordination variability in individuals with flat feet.

This could assist in designing personalized gait retraining programs tailored to individuals with flat feet or pronated foot structures, enhancing long-term outcomes.

## Materials and methods

### Study design

This is a cross-sectional study conducted to determine the effects of wide step width using real-time visual feedback on lower limb inter-joint coordination and coordination variability in individuals with flat feet.

### Setting

Data were collected at the Motion Lab of Mowafaghian, Research Center for Intelligent Neuro-Rehabilitation Technologies in Tehran. Recruitment and testing were conducted from January to March 2024.

### Participants

G*power analysis indicated that 20 subjects are needed to achieve a statistical power of 0.80, assuming an effect size of 0.80 based on rearfoot eversion effect size reported by Mousavi et al. 's study [22] and an alpha level of 0.05. Physically active individuals with pronated feet were recruited by our advertisements and social media from local running clubs to volunteer for participation. Inclusion criteria were male and female rearfoot striker, pronated feet individuals with static navicular drop ≥ 0.9 cm [23] and rearfoot eversion ≤ -4° [24] aged 18–40, engaged in exercise at least three times a week for the past year, with no self-reported lower-limb injuries or pain in the last six months, and free of any musculoskeletal disorders or pain before data collection. Twenty volunteers who met the inclusion criteria participated in this study. Ethical approval was obtained through the local Medical Ethics Committee IR.UT.SPORT.REC.1402.123. Subjects signed an informed written consent form and completed a self-developed questionnaire for demographic information prior to data collection. The recruitment process began on April 3 and concluded on April 23, 2024.

### Instrumentation

Running assessments were performed on a treadmill (S 3301, SPORTEC, Taiwan). Kinematic data were recorded at 120 Hz using the gold standard 10-camera integrated 3D motion capture system (Six IR Cameras: MX T40-S; Four IR Cameras: Vero (V2.2); Vicon Motion Systems, Oxford, UK).

### Marker placement

Fig 1. Shows the marker placement. A total of twenty-six passive reflective markers (14 mm) were affixed to the subject's body by the same investigator (FKH). Of these, 16 markers were attached according to lower body Plug-in-Gate [25] at anterior and posterior superior iliac spine, lateral epicondyle, lower later 1/3 of the thigh and shank, lateral malleolus, second metatarsal head and calcaneus and 8 markers were attached to both feet at the first metatarsal head, navicular bone tuberosity, medial side of calcaneus and posterior part of calcaneus for measuring rearfoot in/eversion and MLAA. Two markers placed along the vertical midpoint of the heel, along with a marker on the medial side of the heel, were used to define the rearfoot segment.

### Baseline measurement

To set a certain running speed for giving real-time feedback and generalize it to all runners, treadmill speed was set at 2.5 km/h for walking and 8 km/h for running in all conditions. A standard running speed of 8 km/h was set for all participants under different conditions, based on findings from pilot testing. This speed which was determined through pilot test, allowed participants to concentrate on adjusting their step width while running, as doing so became more difficult at higher

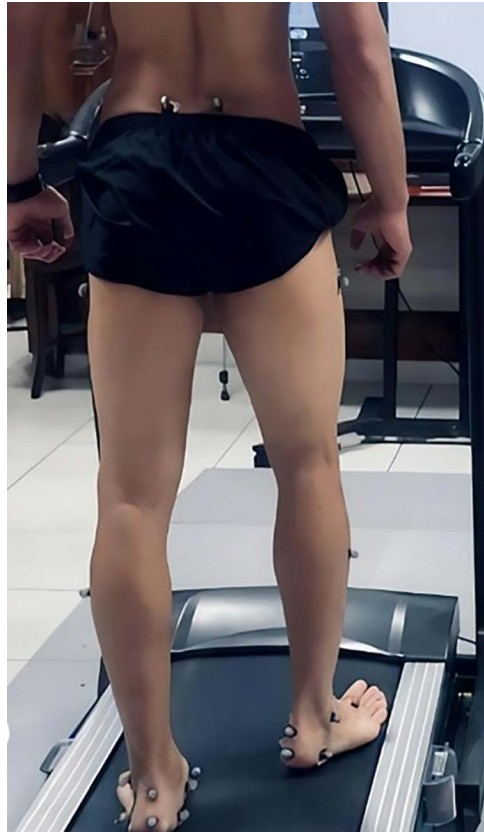

**Fig 1. Gait retraining with biofeedback.**

speeds. All participants completed the conditions without footwear [21]. After warming up for 5 minutes, a 20-second baseline dataset was gathered, consisting of at least 20 strides. The step width, defined as the distance between the right and left heel contacts in the frontal plane, was measured over the first 20 strides to determine the baseline step width.

## Feedback

A MATLAB script was developed to generate real-time feedback for wide condition. Feedback was presented on a screen in front of the treadmill. A shaded area with a red and green line set to step width (cm) was designed to reflect step width during midstance in real time. To execute a wide step, the target range was established at 10 cm greater than the average baseline step width while walking, allowing for a deviation area of ± 3 cm deviation from this point (Red, Fig 2). When the pointer was located within the shaded range, the area became green, otherwise it became red. The pointer was fixed on midstance step width and updated with each step. Before the feedback session, participants practiced their normal gait to familiarize themselves with the feedback. After a 2-minute running session with step width feedback preceding each task, subjects were then asked to keep running, and after 1-min running a 20-second dataset was collected. The order of the experimental tasks was randomized.

## Data analysis and calculation of coordination and coordination variability

The gait cycle was divided into: loading response (0–20% of gait cycle), mid-stance (20–47% of gait cycle), push off (47–60% of gait cycle), and swing phase (60–100% of gait cycle). Data for each stride was time-normalized to 100 data

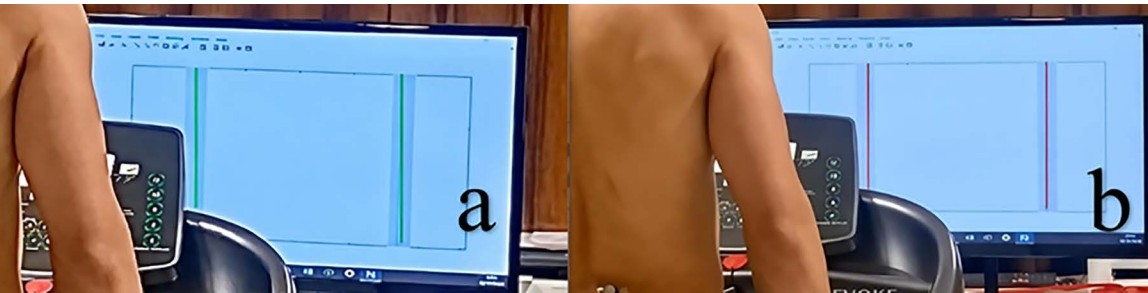

**Fig 2. Picture representing real-time visual feedback for wide step width.** The training process: real-time visual feedback is provided to the subject via the screen. a; The feedback for wide condition when green, b; The feedback for wide condition when red. The aim is to turn the pointer green by keeping the red pointer (step width) inside the shaded area. When the pointer leaves the shaded area, it turns red and when the pointer locates inside the shaded area it turns green.

points using linear interpolation technique [26]. The lowest measure of the heel marker in vertical direction was considered as initial contact and the frame in which the velocity of hallux marker in sagittal plane started to increase was considered as toe-off.

Kinematic data were filtered using NEXUS software a Woltring filter, MSE mode, order 10, and gap filling was based on rigid body and pattern fill. Outcomes of 10 consecutive steps following the given step width were calculated and averaged within subjects before being averaged within conditions. The calculations for coordination and coordination variability were obtained from the studies [4,27,28]. Data was analyzed using a custom MATLAB script (Version R2018a, Natick, MA, USA).

Figs 2 and 3 shows the implications of a coupling angle based on its position in each quadrant, which is essential for understanding the findings presented in Tables.

### Statistical analysis

Shapiro-Wilk test was used to assess normal distribution of data. Comparisons of coordination between normal and wide conditions were conducted using a paired Watson-Williams test designed for circular data. These analyses were performed with Oriana software version 3.21 (Wales, UK). Additionally, within-group coordination variability was assessed with paired t-tests using IBM SPSS version 23 (IBM Corp., Armonk, NY, USA). In line with [29], we opted not to apply Bonferroni corrections for multiple comparisons to avoid significant reductions in statistical power.

For each significant outcome, we assessed the Cohen's d effect size based on pooled SD [30]. A d value of less than 0.50 reflects small effects, a value between 0.50 and 0.80 signifies medium effects, and a d of 0.80 or above represents large effects.

### Results

Twenty adults were included in this study. Normal distribution of data was met for all variables (p>.05). Table 1 shows the characteristics of 20 participants. Normal and wide step width in midstance are presented in Table 2. The average change in step width was 8.78 cm for the wide condition.

### Inter-joint coordination

Table 3. shows coordination angles during walking with normal and wide step width. Significant differences occurred for sagittal ankle-knee coordination during LR, increasing from 66.0 ± 5.7 to 71.5 ± 5.8, showed a change from In-phase pattern to a proximal motion in wide step width. Significant differences occurred for frontal ankle-knee

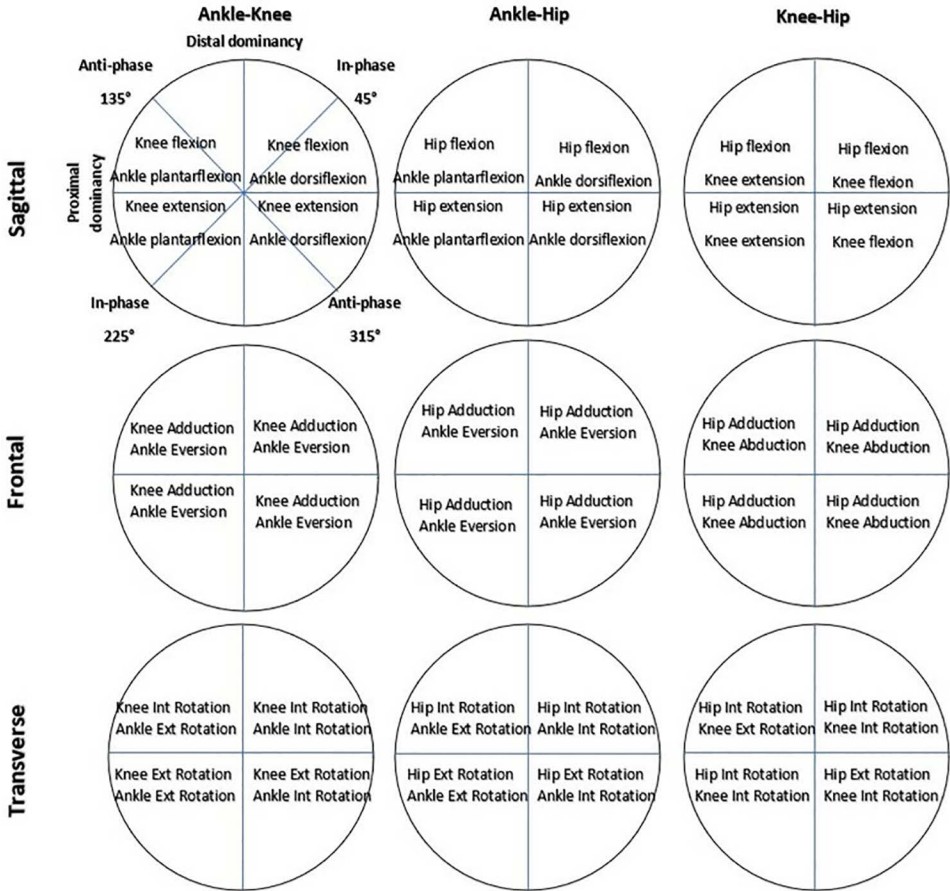

**Fig 3. The correlation between the phase angle in each quadrant and the relative motion of distal and proximal joints is as follows: A coupling angle of 0° or 180° signifies movement in the distal joint without any movement in the proximal joint.** Conversely, a coupling angle of 90° or 270° reflects motion in the proximal joint while the distal joint remains still. Additionally, vector angles of 45°, 135°, 225°, and 315° represent equal relative motion in both the proximal and distal joints.

coordination during mid-stance, decreasing from 55.0 ± 14.0 to 46.8 ± 11.3, indicating a more In-phase motion during wide condition. Significant differences occurred for transverse ankle-hip coordination during PO decreasing from 65.4 ± 9.3 to 52.0 ± 16.4, indicating a more In-phase motion during wide condition. Significant differences occurred for frontal knee-hip coordination during mid-stance decreasing from 50.5 ± 19.4 to 35.7 ± 20, which shows an In-phase pattern with a more hip motion during walking with wide step width. Significant differences occurred for ankle frontal and knee transverse during PO decreasing from 67.2 ± 9.5 to 52.7 ± 18.1, indicating an In-phase pattern during wide walking.

Table 4. shows coordination angles during running normal and wide step width. Significant differences for mean coordination angles during running occurred only for sagittal ankle-knee coordination during LR (Normal (53.4 ± 8.0) and Wide (47.9 ±8.3)) frontal ankle-hip coordination during mid-stance (Normal (62.4 ± 7.0) and Wide (67.2 ± 7.6)) which indicates a more proximal joint motion in wide running) and sagittal knee-hip during LR (Normal (43.1 ± 7.0) and Wide (37.7 ± 6.9)) indicating a more distal joint motion in wide condition) and PO (Normal (40.5 ± 19.6) and Wide (52.6 ± 16.2)) showing an In-phase pattern which changes to a more knee motion in wide running.

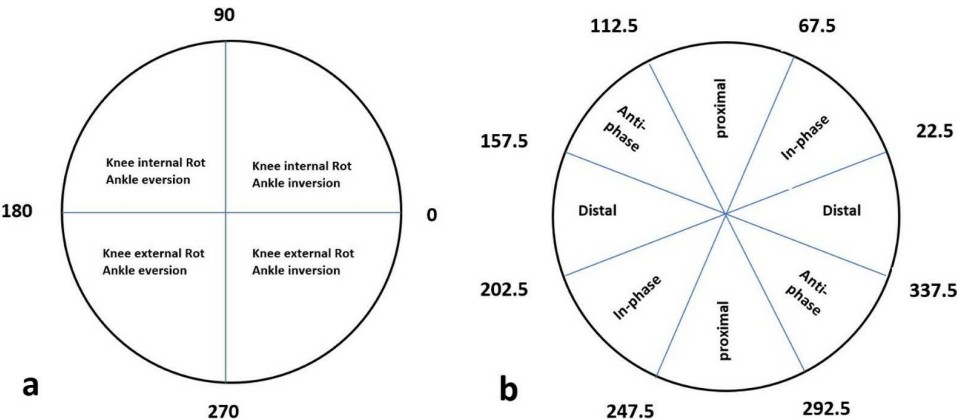

**Fig 4. a; The connection between the phase angle in each quadrant and the relative movements of ankle inversion/eversion and knee internal/external rotation is outlined as follows: A vector angle of 0° or 180° signifies that the ankle is moving independently of the knee.** Conversely, a vector angle of 90° or 270° indicates knee movement occurring without any ankle motion. Furthermore, vector angles of 45°, 135°, 225°, and 315° represent balanced relative motion between the knee and the ankle. Fig 4. b; Areas and angles related to each phase.

**Table 1. Demographics, SD= Standard Deviation.**

| Variable | Mean (SD) | Range |
|---|---|---|
| Age, y | 28.1 (6.11) | 23, 40 |
| Sex | Male: 10, Female: 10 | …. |
| Weight, kg | 72.6 (8.0) | 58, 87 |
| Height, cm | 176 (9.1) | 164, 190 |
| Rearfoot eversion | -6 (3.2) | -4, -11 |
| Navicular Drop | 1.4 (0.4) | 0.9, 2.5 |

**Table 2. Results of normal and wide step width (cm).**

| Wide (walking) | Normal (walking) | Wide (running) | Normal (running) |
|---|---|---|---|
| 18.25 (1.65) | 9.47 (1.2) | 16.65 (2.56) | 8.22 (1.46) |

## Inter-joint coordination variability

Tables 5 and 6. compare coordination variability value between the results obtained in normal condition and wide step width during walking and running respectively. Paired T-test showed significant increase in coordination variability in sagittal ankle-knee coordination during LR ($p<0.001$, d = 0.94) and frontal ankle & transverse knee coordination during PO, and significant decrease in transverse ankle-hip coordination during PO ($p = 0.003$, d = 1.05) between normal and wide conditions during walking.

## Discussion

This study aimed to assess how widening step width affects lower limb inter-joint coordination and its variability in individuals with flat feet. Wide walking showed a shift towards proximal joint motion in sagittal ankle-knee coordination during loading response during LR ($p=0.006$), an In-phase motion in transverse ankle-hip coordination during push-off ($p=0.004$), and an In-phase pattern in frontal knee-hip coordination during mid-stance ($p=0.027$). Frontal ankle and transverse knee

**Table 3. Coordination angle during walking.**

| Coordination angle | Sub-Phase | Normal | Wide | p-value | Effect size |
|---|---|---|---|---|---|
| **Sagittal Ankle-Knee** | LR | 66.0 ± 5.7 | 71.5 ± 5.8 | **0.006** | 0.92 |
| | MS | 15.3 ± 9.5 | 18.3 ± 8.6 | 0.314 | 0.32 |
| | PO | 42.0 ± 12.5 | 38.2 ± 10.4 | 0.308 | 0.92 |
| **Frontal Ankle-Knee** | LR | 57.5 ± 17.7 | 61.0 ± 14.2 | 0.506 | 0.20 |
| | MS | 55.0 ± 14.0 | 46.8 ± 11.3 | **0.05** | 0.62 |
| | PO | 46.7 ± 15.3 | 39.4 ± 23.0 | 0.247 | 0.37 |
| **Transverse Ankle-Knee** | LR | 67.4 ± 7.3 | 67.9 ± 9.6 | 0.833 | 0.06 |
| | MS | 35.2 ± 23.3 | 31.1 ± 17.9 | 0.547 | 0.19 |
| | PO | 44.1 ± 16.4 | 42.6 ± 15.0 | 0.764 | 0.09 |
| **Sagittal Ankle-Hip** | LR | 38.3 ± 9.4 | 36.7 ± 7.5 | 0.571 | 0.18 |
| | MS | 13.0 ± 15.0 | 12.9 ± 13.3 | 0.996 | 0.00 |
| | PO | 60.2 ± 11.9 | 60.3 ± 10.0 | 0.985 | 0.01 |
| **Frontal Ankle-Hip** | LR | 65.4 ± 10.4 | 61.3 ± 14.5 | 0.325 | 0.92 |
| | MS | 45.1 ± 28.6 | 28.5 ± 26.5 | 0.073 | 0.92 |
| | PO | 45.0 ± 17.0 | 44.5 ± 13.0 | 0.904 | 0.04 |
| **Transverse Ankle-Hip** | LR | 64.4 ± 11.2 | 66.5 ± 8.6 | 0.503 | 0.23 |
| | MS | 39.3 ± 27.4 | 34.6 ± 22.7 | 0.568 | 0.17 |
| | PO | 65.4 ± 9.3 | 52.0 ± 16.4 | **0.004** | 1.02 |
| **Sagittal Knee-Hip** | LR | 32.5 ± 10.2 | 36.0 ± 7.2 | 0.226 | 0.39 |
| | MS | 21.7 ± 18.0 | 20.3 14.6 | 0.794 | 0.08 |
| | PO | 38.1 ± 9.6 | 36.1 ± 11.0 | 0.555 | 0.18 |
| **Frontal Knee-Hip** | LR | 55.9 ± 8.4 | 58.2 ± 11.7 | 0.49 | 0.21 |
| | MS | 50.5 ± 19.4 | 35.7 ± 20.1 | **0.027** | 0.71 |
| | PO | 41.0 ± 14.0 | 40.4 ± 17.9 | 0.909 | 0.05 |
| **Transverse Knee-Hip** | LR | 65.5 ± 12.1 | 69.0 ±5.0 | 0.259 | 0.40 |
| | MS | 33.5 ± 24.9 | 34.6 ± 20.5 | 0.876 | 0.04 |
| | PO | 53.9 ± 17.0 | 47.3 ± 18.6 | 0.257 | 0.35 |
| **Ankle frontal - knee transverse** | LR | 65.9 ± 14.2 | 70.4 ± 9.7 | 0.264 | 0.39 |
| | MS | 41.2 ± 27.5 | 42.8 ± 24.1 | 0.847 | 0.06 |
| | PO | 67.2 ± 9.5 | 52.7 ± 18.1 | **0.003** | 1.04 |

coordination during push-off changed to In-phase (p=0.003). Wide running significantly shifted frontal ankle-hip coordination towards proximal joint motion during mid-stance (p=0.05). Transverse ankle-hip coordination showed an in-phase pattern in wide conditions during push-off (p=0.044), during LR (p=0.022). Wide walking, significantly increased coordination variability of the sagittal ankle-knee during LR and decreased transverse ankle-hip during push-off. Wide walking significantly increased coordination variability in ankle-knee in sagittal plane during LR (p<0.001). Wide running significantly decreased the coordination variability in the ankle-knee sagittal during LR (p<0.001) and knee-hip sagittal during LR (p=0.007) and push-off (p=0.016).

## Inter-joint coordination

**Ankle-knee coordination.** Our results indicated that during LR, sagittal plane ankle-knee coordination exhibited a proximal pattern (71°) with wider step width, compared to normal walking. This shift potentially changes gait stability and efficiency [31]. A potential explanation for this shift is that a wider stance may lengthen the mediolateral moment

**Table 4. Coordination angle during running.**

| Coupling angle | Sub-Phase | Normal (°) | Wide (°) | p-value | Effect size |
|---|---|---|---|---|---|
| **Sagittal Ankle-Knee** | LR | 53.4 ± 8.0 | 47.9 ±8.3 | **0.046** | 0.65 |
| | MS | 71.7 ± 3.5 | 70.4 ± 5.4 | 0.377 | 0.29 |
| | PO | 48.8 ± 20.5 | 57.8 ± 15.9 | 0.137 | 0.48 |
| **Sagittal Ankle-Knee** | LR | 56.8 ± 17.5 | 56.4 ± 16.7 | 0.934 | 0.03 |
| | MS | 67.0 ± 10.1 | 70.1 ± 5.0 | 0.232 | 0.41 |
| | PO | 46.1 ± 13.4 | 41.9 ± 10.2 | 0.287 | 0.34 |
| **Frontal Ankle-Knee** | LR | 56.4 ± 12.2 | 50.9 ± 10.5 | 0.139 | 0.48 |
| | MS | 66.3 ± 4.2 | 66.8 ± 6.9 | 0.788 | 0.09 |
| | PO | 12.7 ± 7.3 | 11.7 ± 5.8 | 0.642 | 0.15 |
| **Transverse Ankle-Knee** | LR | 55.8 ± 9.2 | 57.7 ± 8.9 | 0.522 | 0.21 |
| | MS | 63.0 ± 9.8 | 63.6 ± 7.3 | 0.854 | 0.06 |
| | PO | 40.4 ± 24.3 | 39.3 ± 19.4 | 0.883 | 0.05 |
| **Sagittal Ankle-Hip** | LR | 49.8 ± 14.6 | 48.5 ± 20.1 | 0.821 | 0.10 |
| | MS | 62.4 ± 7.0 | 67.2 ± 7.6 | **0.05** | 0.64 |
| | PO | 43.8 ± 19.2 | 39.8 ± 16.6 | 0.495 | 0.21 |
| **Frontal Ankle-Hip** | LR | 59.7 ± 16.4 | 57.3 ± 13.7 | 0.623 | 0.14 |
| | MS | 58.5 ± 16.1 | 68.3 ± 9.0 | 0.025 | 0.78 |
| | PO | 49.5 ± 18.6 | 36.1 ± 28.4 | 0.092 | 0.53 |
| **Transverse Ankle-Hip** | LR | 43.1 ± 7.0 | 37.7 ± 6.9 | **0.022** | 0.76 |
| | MS | 62.7 ± 9.7 | 65.7 ± 6.0 | 0.255 | 0.38 |
| | PO | 40.5 ± 19.6 | 52.6 ± 16.2 | **0.044** | 0.66 |
| **Sagittal Knee-Hip** | LR | 47.8 ± 18.9 | 51.2 ± 15.9 | 0.55 | 0.19 |
| | MS | 62.2 ± 13.7 | 67.2 ± 6.3 | 0.154 | 0.50 |
| | PO | 45.9 ± 17.4 | 37.5 ± 14.2 | 0.112 | 0.51 |
| **Frontal Knee-Hip** | LR | 48.2 ± 16.4 | 46.6 ± 14.7 | 0.764 | 0.09 |
| | MS | 56.9 ± 5.2 | 58.1 ± 9 | 0.501 | 0.21 |
| | PO | 52.2 ± 12.7 | 51.3 ± 17 | 0.852 | 0.07 |
| **Transverse Knee-Hip** | LR | 63.4 ± 18.2 | 62.1 ± 16.9 | 0.817 | 0.07 |
| | MS | 30.1 ± 9.2 | 28.0 ± 10.5 | 0.511 | 0.21 |
| | PO | 60.9 ± 5.8 | 59.8 ± 5.5 | 0.544 | 0.19 |

arm between the body center of mass and center of pressure [32], requiring different muscle activation for balance and propulsion [33]. a wider step may lead to more stable coordination patterns, reducing the risk of malalignment and potential injuries [34]. Moreover, the proximal pattern suggests that the body is using larger, more stable joints to control movement, which can be beneficial in maintaining overall stability during walking [35].

The change in coordination during wider walking may act as a compensatory mechanism to reduce excessive lateral sway, enhancing control over the center of mass and overall stability. In this context, the ankle demonstrated increased lateral movement, reflecting a shift in motion dynamics. During midstance, the ankle and knee transitioned from a knee-dominant motion of 55° in normal walking to a more ankle pattern of 46° with wider step width. This adjustment may indicate a modification in joint movement to meet new biomechanical demands associated with wider stances. While such changes in joint coordination might suggest a potential strategy for enhancing gait mechanics and possibly reducing injury risk [36], further research is needed to directly assess the impact of this new coordination on injury outcomes.

Furthermore, the pattern of ankle frontal and knee transverse shifted to an in-phase pattern (52°) in wider walking compared to a distinct knee internal rotation pattern (67°) in normal walking. Research suggests that individuals with flat

**Table 5. Results of coordination variability during walking.**

| Coordination variability | Sub-Phase | Normal | Wide | p-value | Effect size |
|---|---|---|---|---|---|
| **Sagittal Ankle-Knee** | LR | 66.03 ± 5.88 | 71.47 ± 5.97 | **<0.001** | 0.94 |
| | MS | 15.34 ± 9.81 | 18.33 ± 8.85 | p = 0.351 | 0.32 |
| | PO | 41.95 ± 12.82 | 38.23 ± 10.70 | p = 0.221 | 2.87 |
| **Frontal Ankle-Knee** | LR | 57.26 ± 18.11 | 60.68 ± 14.83 | p = 0.454 | 0.21 |
| | MS | 54.83 ± 14.28 | 46.76 ± 11.57 | p = 0.058 | 0.64 |
| | PO | 46.43 ± 15.77 | 39.15 ± 23.08 | p = 0.133 | 0.38 |
| **Transverse Ankle-Knee** | LR | 67.35 ± 7.52 | 67.88 ± 9.84 | p = 0.793 | 0.062 |
| | MS | 35.11 ± 23.67 | 31.16 ± 18.31 | p = 0.478 | 0.19 |
| | PO | 44.09 ± 16.79 | 42.62 ± 15.19 | p = 0.717 | 0.32 |
| **Sagittal Ankle-Hip** | LR | 38.29 ± 9.68 | 36.74 ± 7.67 | p = 0.300 | 0.18 |
| | MS | 13.44 ± 15.63 | 13.38 ± 14.02 | p = 0.979 | 0.00 |
| | PO | 60.16 ± 12.25 | 60.29 ± 10.10 | p = 0.948 | 0.01 |
| **Frontal Ankle-Hip** | LR | 65.35 ± 10.66 | 61.08 ± 14.98 | p = 0.465 | 0.34 |
| | MS | 44.54 ± 28.90 | 29.08 ± 26.90 | p = 0.015 | 0.57 |
| | PO | 45.06 ± 17.40 | 44.49 ± 12.96 | p = 0.908 | 0.04 |
| **Transverse Ankle-Hip** | LR | 61.08 ± 14.98 | 64.18 ± 11.60 | p = 0.194 | 0.24 |
| | MS | 39.00 ± 27.67 | 34.56 ± 23.04 | p = 0.488 | 0.18 |
| | PO | 65.39 ± 9.57 | 51.88 ± 16.86 | **p = 0.003** | 1.05 |
| **Sagittal Knee-Hip** | LR | 32.45 ±10.42 | 32.97 ± 7.34 | p = 0.300 | 0.40 |
| | MS | 22.02 ± 18.47 | 20.60 ± 15.09 | p = 0.658 | 0.09 |
| | PO | 38.06 ± 9.84 | 36.17 ± 10.96 | p = 0.375 | 0.19 |
| **Frontal Knee-Hip** | LR | 55.90 ± 8.65 | 58.09 ±12.01 | p = 0.465 | 0.22 |
| | MS | 50.23 ± 19.89 | 35.72 ± 20.56 | p = 0.10 | 0.73 |
| | PO | 41.10 ± 13.98 | 40.35 ± 18.31 | p = 0.862 | 0.05 |
| **Transverse Knee-Hip** | LR | 65.39 ± 12.46 | 68.96 ± 5.15 | p = 0.194 | 0.42 |
| | MS | 33.52 ± 25.27 | 34.56 ± 20.91 | p = 0.806 | 0.05 |
| | PO | 53.77 ± 17.44 | 47.31 ± 18.94 | p = 0.253 | 0.36 |
| **Frontal ankle - transverse knee** | LR | 65.45 ± 14.97 | 70.28 ± 10.04 | p = 0.240 | 0.40 |
| | MS | 40.76 ± 27.86 | 42.44 ± 24.57 | p = 0.799 | 0.06 |
| | PO | 67.18 ± 9.72 | 52.42 ± 18.53 | **p = 0.006** | 1.07 |

During running significant decrease in the coordination variability was observed in the ankle-knee sagittal (p<0.001, d = 0.60) in LR and increase in ankle-hip frontal in LR (P = 0.033, d = 0.06), knee-Hip Sagittal in LR (P = 0.007, d = 0.22) and PO (P = 0.016, d = 0.76).

feet, characterized by an increased ankle frontal-knee transverse angle, may be at a higher risk for knee injuries [32]. This highlights the need for further exploration into how step width modifications can influence joint coordination and potentially mitigate injury risk in this population [37].

**Ankle-hip coordination.** Our results revealed that during PO, the transverse ankle-knee coordination shifted significantly from 65˚ to 52˚ in wide compared to normal walking. However, it still has an in-phase pattern. This change may be due to a change of adjustment demands, necessitating for sufficient propulsion.

Additionally, during mid-stance, the frontal plane ankle-hip coordination showed a more proximal pattern in wider walking compared to normal walking. Increased hip adduction, which causes a lateral pelvic shift, may reduce the risk of injuries associated with eversion by moving the CoM closer to the foot's lateral border, promoting ankle inversion [38]. It has been suggested that the hip and subtalar joint work together to manage foot placement and CoM during walking [39], which is crucial for preventing ankle injuries.

**Table 6. Results of coordination variability during running.**

| Coordination variability | Sub-Phase | Normal (°) | Wide (°) | p-value | Effect size |
|---|---|---|---|---|---|
| **Sagittal Ankle-Knee** | LR | 53.48 ± 9.12 | 48.27 ± 8.48 | **<0.001** | 0.60 |
| | MS | 71.84 ± 2.85 | 70.62 ± 6.08 | 0.318 | 0.27 |
| | PO | 51.76 ± 16.95 | 62.20 ± 7.61 | 0.058 | 0.85 |
| **Frontal Ankle-Knee** | LR | 54.08 ± 19.47 | 55.39 ± 19.47 | 0.86 | 0.07 |
| | MS | 64.16 ± 10.48 | 70.10 ± 5.46 | 0.074 | 0.74 |
| | PO | 50.45 ± 9.73 | 45.27 ± 8.42 | 0.077 | 0.57 |
| **Transverse Ankle-Knee** | LR | 56.37 ± 12.12 | 50.57 ± 10.53 | 0.047 | 0.51 |
| | MS | 66.53 ± 3.45 | 66.93 ± 7.79 | 0.86 | 0.07 |
| | PO | 13.81 ± 8.15 | 11.82 ± 6.48 | 0.362 | 0.27 |
| **Sagittal Ankle-Hip** | LR | 58.52 ± 5.64 | 60.02 ± 8.13 | 0.419 | 0.21 |
| | MS | 60.97 ± 9.87 | 62.53 ± 5.89 | 0.574 | 0.20 |
| | PO | 33.39 ± 23.73 | 35.28 ± 21.05 | 0.744 | 0.08 |
| **Frontal Ankle-Hip** | LR | 48.20 ± 15.40 | 48.47 ± 20.29 | 0.972 | 0.01 |
| | MS | 61.22 ± 7.39 | 61.71 ± 8.09 | **0.033** | 0.06 |
| | PO | 50.35 ± 15.07 | 43.93 ± 15.19 | 0.014 | 0.42 |
| **Transverse Ankle-Hip** | LR | 62.76 ± 14.80 | 57.14 ± 13.97 | 0.09 | 0.39 |
| | MS | 54.71 ± 17.56 | 69.75 ± 8.37 | 0.019 | 1.16 |
| | PO | 54.35 ± 12.15 | 38.08 ± 31.02 | 0.09 | 0.75 |
| **Sagittal Knee-Hip** | LR | 58.52 ± 5.64 | 60.02 ± 8.13 | **0.007** | 0.22 |
| | MS | 61.02 ± 10.66 | 65.55 ± 5.81 | 0.208 | 0.55 |
| | PO | 38.69 ± 19.99 | 52.42 ± 16.11 | **0.016** | 0.76 |
| **Frontal Knee-Hip** | LR | 44.20 ± 20.32 | 50.97 ± 17.14 | 0.391 | 0.36 |
| | MS | 58.94 ± 14.93 | 67.69 ± 6.86 | 0.068 | 0.80 |
| | PO | 49.49 ± 16.32 | 37.96 ± 16.10 | 0.096 | 0.71 |
| **Transverse Knee-Hip** | LR | 46.13 ± 16.60 | 46.86 ± 15.91 | 0.86 | 0.04 |
| | MS | 56.37 ± 5.70 | 57.65 ± 6.57 | 0.39 | 0.21 |
| | PO | 54.99 ± 6.30 | 55.63 ± 12.25 | 0.754 | 0.07 |
| **Frontal ankle - transverse knee** | LR | 64.89 ± 19.70 | 59.76 ± 19.58 | 0.116 | 0.26 |
| | MS | 33.16 ± 6.48 | 30.78 ± 8.30 | 0.079 | 0.32 |
| | PO | 59.81± 6.26 | 58.83 ± 6.30 | 0.251 | 0.16 |

**Knee-hip coordination.** Our results showed that during mid-stance, the frontal knee-hip coordination exhibited a more distal in-phase pattern (35°) with wider step widths, compared to 50° in normal walking. While maintaining this in-phase coordination, the movement primarily involved the knee probably for stability.

In addition, during LR and during PO, the sagittal knee-hip coordination indicated a more knee motion (37°) with wider step width, down from 43° in normal running, while still maintaining an in-phase pattern. However, the in-phase coordination is crucial for efficient propulsion and forward movement during gait, regardless of step width.

### Inter-joint coordination variability

**Ankle-knee coordination variability.** There was a significant increase in the sagittal ankle-knee coordination variability during the during LR in wide compared to normal during walking, whereas decreased variability in running could be attributed to participants' concerns about their feet not touching the treadmill borders.

A previous study [40] noted that coordination variability changes with task difficulty when biofeedback is present, and variability has been linked to balance deficits during walking [41], providing insights into neuromotor performance

regulation [42]. Additionally, since shock absorption occurs in the first stance phase, the decreased coordination variability at LR in the wide condition may increase stress on the knee [4].

**Ankle-hip coordination variability.** Decreased coordination variability in the transverse ankle-hip coordination variability during LR was observed in wide compared to normal during walking. Ankle and hip joints must work together to manage foot placement and center of gravity during gait [43]. Two factors influencing joint coordination variability are relative motion and range of motion [44]. The findings suggest that task constraints can limit the variety of potential solutions for achieving movement objectives, resulting in decreased variability.

In contrast, a significant increase in frontal ankle-hip coordination variability was noted during LR while running in wide compared to normal during walking. Modest increases in lower limb variability likely reflect adaptations to the heightened demands of irregular surfaces [45]. Higher coordination variability may indicate greater motor control adaptability to disturbances and sudden changes during movement [46,47]. Increased coordination variability could help alleviate stress on lower limb joints and tissues. Previous studies have linked reduced ankle-hip coordination variability to lower limb injuries, such as ankle sprains [38], patellofemoral pain syndrome [13], and Iliotibial band syndrome [48].

**Knee-hip coordination variability.** During running significant increase was observed in sagittal knee-hip coordination variability in LR and PO. Previous studies highlighted knee-hip coordination and its variability as an important factor for lower limb function [49]. Enhanced knee-hip coordination flexibility in the transverse plane enhances the functionality, aids in attenuating impact forces, and potentially prevents injuries [49].

While no existing research directly compares with our findings, this study provides valuable insights into three-dimensional inter-joint coordination and its variability during walking and running with increased step width in individuals with flexible flat feet. These findings may inform future research, guiding study design and sample size calculations for larger clinical trials involving flexible flat feet and related conditions. Various researchers have emphasized the importance of inter-joint coordination and variability for maintaining dynamic balance and adaptability during gait [50]. Thus, our results may support the inclusion of increased step width in training programs aimed at enhancing inter-joint coordination and variability in individuals with flexible flat feet. This can be implemented in clinical setting by using blocks in a row between feet, using elastic loops around thighs or sticking bands for determining desired step width on the floor.

### Limitations and recommendations

The current study had a few limitations. All participants in the study used a rearfoot strike, so the findings may not be applicable to individuals with midfoot or forefoot strikes. We used the speed of 8 km/h for all participants based on our pilot test, so it is not clear whether the same results would be found at higher or slower speeds. Regarding the type of feedback, our recent study showed that auditory biofeedback may have a more positive effect than visual feedback [51]. Hence, future studies might consider employing real-time auditory feedback instead of visual feedback. Our study'slimitation includes examining results solely in barefoot, thus the intervention'simpact in shod walking/running remains unknown. As our study design, the long-term effects of wide step width on coordination and its variability are unknown.

### Conclusion

The results indicated that wide step width can affect inter-joint coordination during walking/running in flat-footed individuals at certain points. These findings are especially valuable for managing flat-footed individuals as clinicians may incorporate common interventions and gait retraining with wider step width into their treatment plans. Altering step width is a simple, accessible, and non-invasive strategy that can be easily implemented in clinical and everyday settings. Data is provided in S1-S4 files.

## Supporting information

**S1 file. Raw data 1–5.**
(ZIP)

**S2 file. Raw data 6–9 and 15.**
(ZIP)

**S3 file. Raw data 10–14.**
(ZIP)

**S4 file. Raw data 16–20.**
(ZIP)

AcknowledgmentThe present study originates from a research project [21] registered under number 31082.3 at the University of Tehran. The university did not engage in data collection, analysis, reporting, manuscript preparation, or final approval for the publication of research.

## Author contributions

**Conceptualization:** Fateme Khorramroo, Juha M Hijmans, Seyed Hamed Mousavi.

**Data curation:** Fateme Khorramroo, Seyed Hamed Mousavi.

**Formal analysis:** Juha M Hijmans, Seyed Hamed Mousavi.

**Investigation:** Fateme Khorramroo, Seyed Hamed Mousavi.

**Methodology:** Fateme Khorramroo, Juha M Hijmans, Seyed Hamed Mousavi.

**Project administration:** Fateme Khorramroo, Seyed Hamed Mousavi.

**Resources:** Fateme Khorramroo.

**Software:** Fateme Khorramroo, Seyed Hamed Mousavi.

**Supervision:** Juha M Hijmans, Seyed Hamed Mousavi.

**Validation:** Fateme Khorramroo, Juha M Hijmans, Seyed Hamed Mousavi.

**Visualization:** Fateme Khorramroo, Seyed Hamed Mousavi.

**Writing – original draft:** Fateme Khorramroo.

**Writing – review & editing:** Fateme Khorramroo, Juha M Hijmans, Seyed Hamed Mousavi.

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
