## [Decision Letter · Decision Letter 0]

5 Mar 2025

PONE-D-25-00559The Impact of Wide Step Width on Lower Limb Coordination and its Variability in Individuals with flat feetPLOS ONE

Dear Dr. Mousavi,

Thank you for submitting your manuscript to PLOS ONE. After careful consideration, we feel that it has merit but does not fully meet PLOS ONE’s publication criteria as it currently stands. Therefore, we invite you to submit a revised version of the manuscript that addresses the points raised during the review process.

We look forward to receiving your revised manuscript.

Kind regards,

Ateya Megahed Ibrahim El-eglany

Academic Editor

PLOS ONE

Journal Requirements:

4. Please upload a copy of Figure 2, 3, to which you refer in your text on page 7, 8, 9, 10. If the figure is no longer to be included as part of the submission please remove all reference to it within the text.

5. Please remove all personal information, ensure that the data shared are in accordance with participant consent, and re-upload a fully anonymized data set.

Reviewers' comments:

Reviewer's Responses to Questions

**Comments to the Author**

1. Is the manuscript technically sound, and do the data support the conclusions?

Reviewer #1: Yes

Reviewer #2: Partly

2. Has the statistical analysis been performed appropriately and rigorously? 

Reviewer #1: Yes

Reviewer #2: Yes

3. Have the authors made all data underlying the findings in their manuscript fully available?

Reviewer #1: Yes

Reviewer #2: No

4. Is the manuscript presented in an intelligible fashion and written in standard English?

Reviewer #1: Yes

Reviewer #2: Yes

5. Review Comments to the Author

Reviewer #1: The study well well-written and presented because

The study addresses a crucial area in biomechanics and rehabilitation, particularly focusing on individuals with flat feet, which is often under-researched. The paper clearly outlines its objectives and hypotheses, making it easy for readers to understand the purpose and significance of the research. The exploration of wide step width as a factor influencing lower limb coordination presents new insights into gait mechanics that could inform future therapeutic approaches. The inclusion of visual aids, such as graphs and tables, effectively illustrates the data and supports the findings, making the results more accessible to readers.

Some modifications are required:

Abstract:

The methodology part needs more details on measurements

Add the type of study design to the methodology

May you can Add the P-value in the results section of the abstract

The keywords should be according to the MeSh

Introduction:

The study's rationale needs to be explained more, why this study?

Methodology

Check the study be according to the STROBE Checklist

The inclusion and exclusion criteria should be clear

Should be better to have The validity and reliability of the tools

The sample size calculation needs to explain

Add the statistical analysis part (to explain the types of statistics used, The software used, and ….)

Results:

Well describe

Discussion

The first paragraph should mention the objective and main findings,

Explain more the strength and practical implications of the study

Reviewer #2: 1. It is recommended that the research hypothesis be clarified, e.g., “This study hypothesized that wide stride walking guided by visual feedback would reduce rearfoot valgus and improve lower extremity joint coordination in individuals with flat feet.”

2. “This excessive rearfoot eversion is a significant risk factor for running-related injuries, adversely impacting plantar fascia tension and overall foot biomechanics during gait.”, there is a lack of relevant literature to support it. To provide more effective evidence, the authors may consider referring to the following relevant studies: A new method proposed for realizing human gait pattern recognition: Inspirations for the application of sports and clinical gait analysis (DOI: 10.1016/j.gaitpost.2023.10.019) to support the application of gait pattern recognition in sports and clinical gait analysis.

3. In describing the effects of flatfoot on gait, the article uses the terms “lead to” or “result in” several times, but does not distinguish between direct effects (e.g., increased plantar fascia tension due to excessive rearfoot pronation) and indirect effects (e.g., changes in plantar fascia tension (e.g., changes in plantar fascial tension may trigger increased stress on the knee).

4. Clarify the source of the effect sizes (e.g., refer to effect sizes from previous similar studies, or based on data from pilot experiments).

5. Foot biomechanical screening criteria do not detail the source of rearfoot eversion ≤ -4°. Is it based on self-report or validated by kinesiology data?

6. The criterion “pronated feet individuals with static navicular drop > 0.9 cm and rearfoot eversion ≤ -4°” requires more detailed measurements.

7. The statistical analysis section does not indicate whether tests of normality were performed to determine whether appropriate statistical tests (e.g., Watson-Williams test and t-test) were used.

8. The study set a fixed running speed of 8 km/h, but did not provide a rationale for applying this speed to all subjects. Different individuals have different running habits and speed may affect gait adjustment. It is recommended that the rationale for the 8 km/h speed be explained.

9. The study calculated Cohen's d as the effect size, but did not specify which group's standard deviation (e.g., pooled SD or baseline SD) the calculation was based on. It is recommended that the calculation of Cohen's d be clarified to ensure the accuracy of the effect size.

6. PLOS authors have the option to publish the peer review history of their article (what does this mean? ). If published, this will include your full peer review and any attached files.

**Do you want your identity to be public for this peer review?** For information about this choice, including consent withdrawal, please see our Privacy Policy .

Reviewer #1: **Yes: ** Hassan Sadeghi

Reviewer #2: No

---

## [Author Response · Author response to Decision Letter 1]

10 Mar 2025

Dear editor and reviewers;

We thank you for your precious time and insightful and constructive suggestions. Based on your helpful suggestions, we were able to further improve our manuscript. We carefully considered and addressed all your specific comments and revised the text if necessary.

Please find your comments in bold and our responses in italic font. In the manuscript, the highlighted parts in green are the revisions made based on the comments of reviewer1 and the highlighted parts in yellow are the revisions made based on the comments of reviewer2. The comments by reviewer2 which have overlap with the comments from reviewer1 are in green and underlined.

Reviewers' comments:

Reviewer #1

The study well well-written and presented because

The study addresses a crucial area in biomechanics and rehabilitation, particularly focusing on individuals with flat feet, which is often under-researched. The paper clearly outlines its objectives and hypotheses, making it easy for readers to understand the purpose and significance of the research. The exploration of wide step width as a factor influencing lower limb coordination presents new insights into gait mechanics that could inform future therapeutic approaches. The inclusion of visual aids, such as graphs and tables, effectively illustrates the data and supports the findings, making the results more accessible to readers.

Thank you for your positive feedback. It motivated us for improving the manuscript in the best possible way.

Some modifications are required:

Abstract:

The methodology part needs more details on measurements

Thank you for your comment. We added more details to Methods section as bellow:

Twenty flat-footed individuals participated in this cross-sectional study. Lower limb kinematics were assessed by 3-dimensional motion analysis during walking and running on a treadmill with preferred and wide step widths while receiving visual feedback. Inter-joint coordination was quantified using vector-coding for joint angles in the hip, knee, and ankle.

Add the type of study design to the methodology

Thank you for your comment. Study design is provided in the first line of Methods part of Abstarct as shown in the copied part:

Twenty flat-footed individuals participated in this cross-sectional study.

May you can Add the P-value in the results section of the abstract

Thank you for your comment. P-values are added as provided bellow:

Wide walking showed a shift towards proximal joint motion in sagittal ankle-knee coordination during loading response during LR (p=0.006), an In-phase motion in transverse ankle-hip coordination during push-off (p=0.004), and an In-phase pattern in frontal knee-hip coordination during mid-stance (p=0.027). Frontal ankle and transverse knee coordination during push-off changed to In-phase (p=0.003). Wide running significantly shifted frontal ankle-hip coordination towards proximal joint motion during mid-stance (p=0.05). Transverse ankle-hip coordination showed an in-phase pattern in wide conditions during push-off (p=0.044), during LR (p=0.022). Wide walking, significantly increased coordination variability of the sagittal ankle-knee during LR and decreased transverse ankle-hip during push-off. Wide walking significantly increased coordination variability in ankle-knee in sagittal plane during LR (p<0.001). Wide running significantly decreased the coordination variability in the ankle-knee sagittal during LR (p<0.001) and knee-hip sagittal during LR (p=0.007) and push-off (p=0.016).

The keywords should be according to the MeSh

Thank you for your comment. We believe this will improve retrievability of our article. The keywords are changed as the following:

Keywords: Foot, Pronation, gait, Gait Analysis, Biomechanical, Walking, Running, Vector coding, Rearfoot eversion

Introduction:

The study's rationale needs to be explained more, why this study?

Thank you for your comment. This is how we changed the last paragraph of Introduction:

Modifying step width to address rearfoot eversion has emerged as a promising approach beside other interventions; i.e. changing speed, foot progression angle or center of pressure [18], soft tissue mobilization and electrotherapy [19,20]. Evidence indicating that wider step widths can effectively reduce rearfoot eversion during walking and running [18,21]. However, the impact of this modification on lower limb inter-joint coordination has yet to be thoroughly investigated. Understanding how variations in step width influence joint coordination in individuals with flat feet is crucial for elucidating the adaptive mechanisms and developing strategies to optimize gait mechanics in this population. This suggests potential applications in injury prevention and rehabilitation for conditions like patellofemoral pain syndrome or tibial stress injuries [11]. Therefore, the purpose of this study is to examine the effects of wide step width through visual feedback on lower extremity joint coordination and its variability in individuals with flat feet. We hypothesized that wide stride gait guided by visual feedback would reduce rearfoot eversion and improve lower extremity joint coordination in individuals with flat feet. This could assist in designing personalized gait retraining programs tailored to individuals with flat feet or pronated foot structures, enhancing long-term outcomes.

Methodology

Check the study be according to the STROBE Checklist

Thank you for your comment. We provided the filled checklist with our response and revide manuscript.

The inclusion and exclusion criteria should be clear

Thank you for your comment. We added reference to the study we chose the inclusion and exclusion criteria from:

G*power analysis indicated that 20 subjects are needed to achieve a statistical power of 0.80, assuming an effect size of 0.80 based on rearfoot eversion effect size reported by Mousavi et al. ‘s study [22] and an alpha level of 0.05. Physically active individuals with pronated feet were recruited by our advertisements and social media from local running clubs to volunteer for participation. Inclusion criteria were male and female rearfoot striker, pronated feet individuals with static navicular drop > 0.9 cm [23] and rearfoot eversion ≤ -4° [24] aged 18–40, engaged in exercise at least three times a week for the past year, with no self-reported lower-limb injuries or pain in the last six months, and free of any musculoskeletal disorders or pain before data collection. Twenty volunteers who met the inclusion criteria participated in this study. Ethical approval was obtained through the local Medical Ethics Committee IR.UT.SPORT.REC.1402.123. Subjects signed an informed written consent form and completed a self-developed questionnaire for demographic information prior to data collection. The recruitment process began on April 3 and concluded on April 23, 2024.

23. Cote KP, Brunet ME, Gansneder BM, Shultz SJ. Effects of Pronated and Supinated Foot Postures on Static and Dynamic Postural Stability. Journal of athletic training. 2005;40: 41–46.

24. Bok S-K, Kim B-O, Lim J-H, Ahn S-Y. Effects of custom-made rigid foot orthosis on pes planus in children over 6 years old. Annals of rehabilitation medicine. 2014;38: 369–375. doi:10.5535/arm.2014.38.3.369

Should be better to have The validity and reliability of the tools

Thank you for your comment. The changed parts are shown bellow:

Running assessments were performed on a treadmill (S 3301, SPORTEC, Taiwan). Kinematic data were recorded at 120 Hz using the gold standard 10-camera integrated 3D motion capture system (Six IR Cameras: MX T40-S; Four IR Cameras: Vero (V2.2); Vicon Motion Systems, Oxford, UK).

The sample size calculation needs to explain

Thank you. We provided information as bellow:

Setting

Data were collected at the Motion Lab of Mowafaghian, Reasearch Center for Intelligent Neuro-Rehabilitation Technologies in Tehran. Recruitment and testing were conducted from January to March 2024.

Participants

G*power analysis indicated that 20 subjects are needed to achieve a statistical power of 0.80, assuming an effect size of 0.80 based on rearfoot eversion effect size reported by Mousavi et al. ‘s study [22] and an alpha level of 0.05….

Add the statistical analysis part (to explain the types of statistics used, The software used, and ….)

Thank you for your comment. There already was a statistical part. We provide the edited version here:

Shapiro-Wilk test was used to assess normal distribution of data. Comparisons of coordination between normal and wide conditions were conducted using a paired Watson-Williams test designed for circular data. These analyses were performed with Oriana software version 3.21 (Wales, UK). Additionally, within-group coordination variability was assessed with paired t-tests using IBM SPSS version 23 (IBM Corp., Armonk, NY, USA). In line with [28], we opted not to apply Bonferroni corrections for multiple comparisons to avoid significant reductions in statistical power.

For each significant outcome, we assessed the Cohen’s d effect size based on pooled SD [29]. A d value of less than 0.50 reflects small effects, a value between 0.50 and 0.80 signifies medium effects, and a d of 0.80 or above represents large effects.

Results:

Well describe

Thank you for your positive feedback

Discussion

The first paragraph should mention the objective and main findings,

Thank you for your helpful suggestion. We copied the first paragraph here:

This study aimed to assess how widening step width affects lower limb inter-joint coordination and its variability in individuals with flat feet. Wide walking showed a shift towards proximal joint motion in sagittal ankle-knee coordination during loading response during LR(p=0.006), an In-phase motion in transverse ankle-hip coordination during push-off (PO)(p=0.004), and an In-phase pattern in frontal knee-hip coordination during mid-stance (p=0.027). Frontal ankle and transverse knee coordination (PO) changed to In-phase (p=0.003). Wide running significantly shifted frontal ankle-hip coordination towards proximal joint motion (mid-stance)(p=0.05). Transverse ankle-hip coordination showed an in-phase pattern in wide conditions (PO)(p=0.044), (LR)(p=0.022). Wide walking, significantly increased coordination variability of the sagittal ankle-knee (LR) and decreased transverse ankle-hip (PO). Wide walking significantly increased coordination variability in ankle-knee in sagittal plane (LR)(p<0.001). Wide running significantly decreased the coordination variability in the ankle-knee sagittal (LR)(p<0.001) and knee-hip sagittal (LR)(p=0.007), (PO)(p=0.016).

Explain more the strength and practical implications of the study

Thank you for your comment. We added the following highlighted parts:

While no existing research directly compares with our findings, this study provides valuable insights into three-dimensional inter-joint coordination and its variability during walking and running with increased step width in individuals with flexible flat feet. These findings may inform future research, guiding study design and sample size calculations for larger clinical trials involving flexible flat feet and related conditions. Various researchers have emphasized the importance of inter-joint coordination and variability for maintaining dynamic balance and adaptability during gait [49]. Thus, our results may support the inclusion of increased step width in training programs aimed at enhancing inter-joint coordination and variability in individuals with flexible flat feet. This can be implemented in clinical setting by using blocks in a roe between feet, using elastic loops around thighs or sticking bands for determining desired step width on the floor.

Conclusion

The results indicated that wide step width can affect interjoint coordination during walking/running in flat-footed individuals at certain points. These findings are especially valuable for managing flat-footed individuals as clinicians may incorporate common interventions and gait retraining with wider step width into their treatment plans. Altering step width is a simple, accessible, and non-invasive strategy that can be easily implemented in clinical and everyday settings.

Reviewer #2

1. It is recommended that the research hypothesis be clarified, e.g., “This study hypothesized that wide stride walking guided by visual feedback would reduce rearfoot valgus and improve lower extremity joint coordination in individuals with flat feet.”

Thank you for your comment. We edited the hypothesis as bellow:

We hypothesized that wide stride gait guided by visual feedback would reduce rearfoot eversion and improve lower extremity joint coordination in individuals with flat feet.

2. “This excessive rearfoot eversion is a significant risk factor for running-related injuries, adversely impacting plantar fascia tension and overall foot biomechanics during gait.”, there is a lack of relevant literature to support it. To provide more effective evidence, the authors may consider referring to the following relevant studies: A new method proposed for realizing human gait pattern recognition: Inspirations for the application of sports and clinical gait analysis (DOI: 10.1016/j.gaitpost.2023.10.019) to support the application of gait pattern recognition in sports and clinical gait analysis.

Thnk you for your comment. This is how we edited and referenced this part:

Prolonged pronation, particularly caused by excessive rearfoot eversion, commonly seen in flat feet, can lead to compensatory changes in tibial [8] and hip rotation [9], resulting in a greater knee valgus angle due to the interconnected movements between rearfoot inversion/eversion and tibial rotation and low back pain [10]. This excessive rearfoot eversion is a significant risk factor for running-related injuries [11], adversely impacting plantar fascia tension and overall foot biomechanics during gait [12]. These compensatory mechanisms can further disrupt the lower extremities joints coordination [4] and potentially lead to increased plantar fascia tention due to excessive rearfoot pronation [12], may trigger increased stress on the knee. Moreover, coordination between adjacent segments has been implicated in the development of injuries such as iliotibial band syndrome [13].

12. Xu D, Zhou H, Quan W, Jiang X, Liang M, Li S, et al. A new method proposed for realizing human gait pattern recognition: Inspirations for the application of sports and clinical gait analysis. Gait & posture. 2024;107: 293–305. doi:10.1016/j.gaitpost.2023.10.019

3. In describing the effects of flatfoot on gait, the article uses the terms “lead to” or “result in” several times, but does not distinguish between direct effects (e.g., increased plantar fascia tension due to excessive rearfoot pronation) and indirect effects (e.g., changes in plantar fascia tension (e.g., changes in plantar fascial tension may trigger increased stress on the knee).

Thank you for your constructive suggestion. The way we edited this section is shown in response to your comment number 2.

4. Clarify the source of the effect sizes (e.g., refer to effect sizes from previous similar studies, or based on data from pilot experiments).

Thnk you for your comment. We provided it as bellow:

G*power analysis indicated that 20 subjects are needed to achieve a statistical power of 0.80, assuming an effect size of 0.80 based on rearfoot eversion effect size reported by Mousavi et al. ‘s study [22] and an alpha level of 0.05.

For each significant outcome, we assessed the Cohen’s d effect size based on pooled SD [29]. A d value of less than 0.50 reflects small effects, a value between 0.50 and 0.80 signifies medium effects, and a d of 0.80 or above represents large effects.

5. Foot biomechanical screening criteria do not detail the source of rearfoot eversion ≤ -4°. Is it based on self-report or validated by kinesiology data?

Thank you for your helpful comment. We used the method by Bok et al. ‘s method which we referenced now.

Physically active individuals with pronated feet were recruited by our advertisements and social media from local running clubs to volunteer for participation. Inclusion criteria were male and

---

## [Decision Letter · Decision Letter 1]

14 Mar 2025

The Impact of Wide Step Width on Lower Limb Coordination and its Variability in Individuals with flat feet

PONE-D-25-00559R1

Dear AuthorsWe pleased to inform you that your manuscript has been judged scientifically suitable for publication and will be formally accepted for publication once it meets all outstanding technical requirements.

Kind regards,

Ateya Megahed Ibrahim El-eglany

Academic Editor

PLOS ONE

Additional Editor Comments (optional):

Reviewers' comments:

Reviewer's Responses to Questions

**Comments to the Author**

1. If the authors have adequately addressed your comments raised in a previous round of review and you feel that this manuscript is now acceptable for publication, you may indicate that here to bypass the “Comments to the Author” section, enter your conflict of interest statement in the “Confidential to Editor” section, and submit your "Accept" recommendation.

Reviewer #1: All comments have been addressed

Reviewer #2: (No Response)

2. Is the manuscript technically sound, and do the data support the conclusions?

Reviewer #1: Yes

Reviewer #2: (No Response)

3. Has the statistical analysis been performed appropriately and rigorously? 

Reviewer #1: Yes

Reviewer #2: (No Response)

4. Have the authors made all data underlying the findings in their manuscript fully available?

Reviewer #1: Yes

Reviewer #2: (No Response)

5. Is the manuscript presented in an intelligible fashion and written in standard English?

Reviewer #1: Yes

Reviewer #2: (No Response)

6. Review Comments to the Author

Reviewer #1: The paper has significantly improved, as the authors diligently addressed all the comments and suggestions from the reviewers. Their thorough revisions not only enhanced the overall clarity and coherence of the content but also elevated the quality of the research, making it more valuable and impactful within the field.

Reviewer #2: All comments have been addressed.

7. PLOS authors have the option to publish the peer review history of their article (what does this mean? ). If published, this will include your full peer review and any attached files.

**Do you want your identity to be public for this peer review?** For information about this choice, including consent withdrawal, please see our Privacy Policy .

Reviewer #1: No

Reviewer #2: No

---

## [Editor Report · Acceptance letter]

PONE-D-25-00559R1

PLOS ONE

Dear Dr. Mousavi,

I'm pleased to inform you that your manuscript has been deemed suitable for publication in PLOS ONE. Congratulations! Your manuscript is now being handed over to our production team.

Kind regards,

on behalf of

Dr. Ateya Megahed Ibrahim El-eglany

Academic Editor

PLOS ONE